# Mechanisms Underlying the Therapeutic Effects of JianPiYiFei II Granules in Treating COPD Based on GEO Datasets, Network Pharmacology, Molecular Docking, and Molecular Dynamics Simulations

**DOI:** 10.3390/biology13090711

**Published:** 2024-09-11

**Authors:** Liyuan Pang, Yongjuan Zhao, Yang Xu, Chencheng Gao, Chao Wang, Xiao Yu, Fang Wang, Kan He

**Affiliations:** 1Department of Pharmacology, College of Basic Medical Sciences, Jilin University, Changchun 130021, China; pangly22@mails.jlu.edu.cn (L.P.); yxu22@mails.jlu.edu.cn (Y.X.); 2Department of Pulmonary and Critical Care Medicine, China-Japan Union Hospital of Jilin University, Changchun 130021, China; zyj1987@jlu.edu.cn; 3Department of Pathogen Biology, College of Basic Medical Sciences, Jilin University, Changchun 130021, China; ccgao22@mails.jlu.edu.cn (C.G.); wangc19@mails.jlu.edu.cn (C.W.); 4Department of Histology & Embryology, College of Basic Medical Sciences, Jilin University, Changchun 130021, China; yuxiao23@mails.jlu.edu.cn

**Keywords:** JPYF II granules, COPD, network pharmacology, molecular docking, molecular dynamics simulation

## Abstract

**Simple Summary:**

COPD is a lung disease characterized by limited respiratory airflow aggravated with time. JPYF II granules are a traditional Chinese medicine used in the treatment of COPD. However, the main components and potential mechanisms of JPYF II granules are still unclear. The purpose of this study was to elucidate the potential mechanism underlying its ability to treat COPD through network pharmacology, molecular docking, and molecular dynamics simulation techniques. Kaempferol, quercetin, and stigmasterol are the main active compounds in the JPYF II Formula in the treatment of COPD, and AKT1, IL-6, and TNF are potential target proteins for the JPYF II Formula in the treatment of COPD. The potential effective compounds, targets, and related potential molecular mechanisms obtained here provide a reference for follow-up studies on COPD.

**Abstract:**

Background: JianPiYiFei (JPYF) II granules are a Chinese medicine for the treatment of chronic obstructive pulmonary disease (COPD). However, the main components and underlying mechanisms of JPYF II granules are not well understood. This study aimed to elucidate the potential mechanism of JPYF II granules in the treatment of COPD using network pharmacology, molecular docking, and molecular dynamics simulation techniques. Methods: The active compounds and corresponding protein targets of the JPYF II granules were found using the TCMSP, ETCM, and Uniport databases, and a compound–target network was constructed using Cytoscape3.9.1. The COPD targets were searched for in GEO datasets and the OMIM and GeneCards databases. The intersection between the effective compound-related targets and disease-related targets was obtained, PPI networks were constructed, and GO and KEGG enrichment analyses were performed. Then, molecular docking analysis verified the results obtained using network pharmacology. Finally, the protein–compound complexes obtained from the molecular docking analysis were simulated using molecular dynamics (MD) simulations. Results: The network pharmacological results showed that quercetin, kaempferol, and stigmasterol are the main active compounds in JPYF II granules, and AKT1, IL-6, and TNF are key target proteins. The PI3K/AKT signaling pathway is a potential pathway through which the JPYF II granules affect COPD. The results of the molecular docking analysis suggested that quercetin, kaempferol, and stigmasterol have a good binding affinity with AKT1, IL-6, and TNF. The MD simulation results showed that TNF has a good binding affinity with the compounds. Conclusions: This study identified the effective compounds, targets, and related underlying molecular mechanisms of JPYF II granules in the treatment of COPD through network pharmacology, molecular docking, and MD simulation techniques, which provides a reference for subsequent research on the treatment of COPD.

## 1. Introduction

COPD is an inflammatory pulmonary syndrome that worsens over time and is characterized by limited respiratory airflow [1]. Inhalation of harmful particles, such as tobacco smoke and pollutants, is the major cause of COPD. Other factors include genetics, age, and social factors [2]. According to reports, COPD causes high mortality and is predicted to become the third leading cause of death in the world by 2030 [3]. Patients with COPD will suffer huge economic losses due to their long treatment cycles and impaired working ability due to physical problems. Therefore, COPD is a healthcare issue of concern [4,5]. Despite this, COPD is a lung disease that can be prevented and treated. Clinically, smoking cessation is the most effective treatment for COPD for smokers. Some medications can also be taken to relieve the symptoms of COPD and prevent complications. Traditional Chinese medicine (TCM) formulations are mostly composed of complex components and have therapeutic effects on COPD [6]. Hence, it is of practical significance to understand the mechanisms of traditional Chinese medicines in the treatment of COPD.

Clinical trials have shown that JPYF II granules have a good improvement effect in the treatment of COPD patients. Since 2008, the Guangdong Provincial Hospital of Traditional Chinese Medicine has used JPYF II granules for the treatment of COPD (CHICCTR-IOR-16008082). JPYF II granules are composed of eight types of traditional Chinese medicine ingredients, including Radix Astragali (HQ), Radix Codonopsis (DS), Radix Bupleuri (CH), Rhizoma Atractylodis Macrocephalae (BZ), Rhizoma Cimicifugae (SM), Fructus Viticis Negundinis (MJZ), Herba Cynomorii (SY), and Semen Persicae (TR), which are shown in Figure 1 [7,8]. Related pharmacological studies have shown that JPYF II granules play a role in the treatment of COPD by inhibiting the reactive oxygen species–endoplasmic reticulum stress–Ca^2+^ signaling pathway and through the NF-κB signaling pathway [9,10].

Network pharmacology is an academic discipline that studies the mechanism of disease and drug action in the context of biological networks, aiming to elucidate the relationship between drug components, diseases, and targets [11,12,13]. Due to the complexity of diseases and TCM ingredients, precise and effective therapeutic interventions, scientific and effective interpretation mechanisms, and accelerated clinical translation can be realized through network pharmacology [14,15,16]. In this study, the targets of JPYF II granules and their active compounds were constructed and analyzed by network pharmacology. Molecular docking technology is now used in modern drug design to explore the conformation of intermolecular binding sites, as well as for drug screening and to observe ligand–target interactions at the molecular level [17,18,19]. MD simulation is a molecular simulation method based on classical mechanics. It is often used to simulate biological macromolecule systems to understand the dynamic processes in biological systems [20].

This research adopted molecular docking, network pharmacology, and MD simulation methods, aiming to elucidate the main components and molecular mechanism of JPYF II granules in the treatment of COPD and provide directions and ideas to obtain an in-depth understanding of diseases and aid in the development of new drugs. A flowchart of this research is shown in Figure 2.

## 2. Materials and Methods

### 2.1. Network Pharmacology

#### 2.1.1. Screening for Bioactive Compounds and Targets from JianPiYiFei II Formula

The bioactive compounds and relative targets of the herbs found in the JianPiYiFei II Formula were collected from the Traditional Chinese Medicine Systems Pharmacology Database (TCMSP) (accessed on 1 June 2023) (https://old.tcmsp-e.com/tcmsp.php) [21] and The Encyclopedia of Traditional Chinese Medicine (accessed on 2 June 2023) (ETCM) (http://www.tcmip.cn/ETCM/) [22]. Oral bioavailability (OB) refers to the speed and degree at which the drug in the preparation is absorbed into the human circulatory system. Drug similarity (DL) represents the similarity of a compound to an available drug. The main bioactive components of the JPYF II Formula and the corresponding target proteins were obtained by OB ≥ 30, DL ≥ 0.18 [23,24]. The obtained target protein was screened as “Reviewed” and “human” to obtain a unique corresponding gene name by using the Uniport database (accessed on 4 June 2023) (https://www.UniProt.org/) [25]. The selected bioactive compounds and their corresponding genes were organized in Excel 2016 sheets.

#### 2.1.2. Construction of Compound–Target Networks

Cytoscape is the software that was used for open-source network visualization and analysis [26]. Its core lies in the network. A simple network diagram consists of nodes and edges. The active compounds and their corresponding targets obtained after the screening of the JPYF II Formula were input into Cytoscape 3.9.1, and then the compound and target network of the JPYF II Formula was constructed. By analyzing the network diagram and making judgments based on the degree value, the main compound components can be visually screened out [27].

#### 2.1.3. Identification of Potential Targets of COPD

The mRNA expression maps of different series of COPD patients and normal patients (from GSE103174 and GSE29133) were downloaded via the Gene Expression Omnibus datasets (accessed on 10 June 2023) at GEO: www.ncbi.nlm.nih.gov/geo/. Then, the R package “limma” was utilized to identify the differentially expressed genes (DEGs) with the criteria of |logFC| > 1 and P < 0.01 [28]. Heatmaps and volcano plots were obtained by using the pheatmap packages and ggplot2 in R (2022.07.2). The COPD-related targets were screened by the GeneCards database (accessed on 15 June 2023) (https://www.genecards.org) [29] and the Online Mendelian Inheritance in Man (accessed on 15 June 2023) (OMIM, https://www.omim.org) [30]. Using COPD as the keyword, the two databases were searched. Then, the obtained targets were deduplicated in preparation for subsequent studies.

#### 2.1.4. Construction of PPI Network

The drugs’ disease targets and other targets were imported into the online tool, Venny (accessed on 17 June 2023) (https://bioinfogp.cnb.csic.es/tools/venny/) [31], to obtain overlapping targets of the two. The overlapping targets were imported into the string database (accessed on 19 June 2023) (https://string-db.org/) [32], and the input protein–protein interaction (PPI) network diagram was obtained by using “Homo sapiens” as the filter condition. The data were downloaded and imported into Cytoscape 3.9.1, and then the CytoNCA plugin [33] was used; betweenness centrality (BC) was used as a metric to screen out the core genes.

#### 2.1.5. GO Enrichment and KEGG Pathway Analyses

We utilized the DAVID database (accessed on 21 June 2023) (https://david.ncifcrf.gov/) [34] to collect the GO analysis data for analyzing the biological function of the targets in COPD. According to the count size, the results of the top 10 biological processes (BP), cellular components (CC), and molecular function (MF) are screened, visualized, and mapped through online website tools (accessed on 21 June 2023) (www.bioinformatics.com.cn) [35]. KEGG pathway enrichment analysis is designed to obtain important signaling pathways. It also adopts the online database Metascape (accessed on 21 June 2023) (https://metascape.org) [36] and uses “human sapiens” as the filter condition for visual enrichment analysis.

### 2.2. Molecular Docking

Molecular docking calculations involve searching for the ligand–acceptor interaction and the optimal binding mode between them by following the principles of spatial structure complementarity and energy minimization in the active site region of the receptor. The three-dimensional structure of the protein encoded by the top 3 core target gene proteins is downloaded from the Protein Data Bank (PDB) database (accessed on 5 July 2023) (https://www.rcsb.org/) [37]. The protein molecules are processed by PyMol to eliminate water molecules and ligands [38] (Version 2.5). We downloaded the 2D structure of the top three active pharmaceutical compounds from the PubChem database (accessed on 6 July 2023) (http://pubchem.ncbi.nlm.nih.gov) [39] and used chem3D [40] (Version 20.0) for energy minimization and format conversion. Molecular docking was conducted via the AutoDockTools software [41] (Version 1.5.7) and visualization via the PyMol (Version 2.5) software and LigPlot+ versions 2.2 [42].

### 2.3. Molecular Dynamics Simulation

Based on the negative binding energy from the molecular docking results, we selected the Desmond section of Schrödinger to further explore the stability of protein–ligand complexes and conduct MD simulation studies. To analyze the stability and conformational behavior of the protein–ligand complex within 100 ns, the following process was simulated. Firstly, the T1P3P system was selected for the system’s construction. For the neutralization stability of the solution system, salt particles should be added according to the specific conditions to simulate the ion concentration in the real biological environment. Secondly, energy minimization was performed to reduce the atomic collisions. Finally, the formal MD simulation was carried out, and the corresponding simulation results, such as RMSD, RMSF, and the protein–ligand interaction, were obtained [20,43,44].

## 3. Results

### 3.1. Screening of Active Compounds for JianPiYiFei II Granules and Potential Targets

Using the TCMSP database with an OB ≥ 30 and DL ≥ 0.18 as the filtering criteria in combination with the ETCM database, 317 active compounds of CH, 195 active compounds of DS, 20 active compounds of BZ, 420 active compounds of HQ, 34 active compounds of MJZ, 77 active compounds of SM, 38 active compounds of SY, and 107 active compounds of TR were obtained. Stigmasterol is a common active compound found in CH, DS, and SM. Isorhamnetin, kaempferol, and quercetin are active compounds shared by CH and HQ. Hederagenin is an active compound shared by HQ and TR; (3S,8S,9S,10R,13R,14S,17R)-10,13-dimethyl-17-[(2R,5S)-5-propan-2-yloctan-2-yl]-2,3,4,7,8,9,11,12,14,15,16,17-dodecahydro-1H-cyclopenta[a]phenanthren-3-ol is the common active compound for HQ and BZ. Beta-sitosterol is an active compound shared by MJZ and TR. These common components are presented in Table 1. Additionally, up to 231 drug target gene proteins were added through the Uniport database after duplicate removal.

### 3.2. Compound–Target Network

Traditional Chinese medicine compounds have multi-target and multi-component characteristics. We utilized the software Cytoscape 3.9.1 to create the interaction network diagram of active compounds and drug target gene proteins and visualized 313 nodes and 1290 edges, which are shown in Figure 3. For the JPYF II Formula’s eight ingredients of Chinese medicine, through the analysis of the network diagram, the three active compounds with the highest scores were kaempferol (with a degree of 292), quercetin (with a degree of 116), and stigmasterol (with a degree of 84). This information is listed in Table 2.

### 3.3. COPD-Related Genes

Based on the expression levels in two series (GSE103174 and GSE29133) of samples from COPD patients and normal patients, 332 DEGs were obtained, among which there were 183 up-regulated genes and 149 down-regulated genes. Heatmaps of some DEGs and their expression patterns are also presented. (Figure 4A,C). Additionally, by using “COPD” as the search term, related disease genes were obtained from the OMIM and Genecards databases. After removing the duplicates, a total of 2096 related genes were obtained. The above 2428 genes were used for subsequent analysis.

### 3.4. PPI Network

The drug-related target genes and the disease-related target genes were obtained via the Venny database, and 156 common genes were acquired (Figure 4B). Subsequently, these common genes were imported into the String database, and a protein–protein interaction network was constructed with “Homo sapiens” as the screening condition (Figure 5a). The PPI network diagram was further analyzed in Cytoscape 3.9.1. The core gene was obtained by using the CytoNca plugin, and the top three genes, namely AKT1, IL-6, and TNF, were selected for subsequent analysis (Figure 5b).

### 3.5. GO Enrichment Analysis

The GO enrichment analysis of 156 overlapping target gene proteins generated a total of 1046 GO items. There are 149 terms related to molecular function (MF), 90 terms related to cellular component (CC), and 785 terms related to biological process (BP). According to the P-value magnitude and the number of enriched genes, the top terms are as follows: positive regulation of transcription from RNA polymerase II promoter, negative regulation of apoptotic process, positive regulation of transcription, DNA-templated, positive regulation of gene expression, apoptotic process, signal transduction, response to xenobiotic stimulus, inflammatory response, positive regulation of cell proliferation, and negative regulation of transcription from RNA polymerase II promoter (Figure 6a).

### 3.6. KEGG Pathway Enrichment Analysis

The KEGG enrichment analysis of 156 overlapping target gene proteins shows the role of the pathway in the pathological mechanism. The 20 most significant signal pathways are presented in a histogram (Figure 6b), which involves the PI3K/AKT signaling pathway, FoxO signaling pathway, JAK-STAT signaling pathway, etc. However, the enrichment results of the PI3K/AKT pathway are more significant.

### 3.7. Molecular Docking Evaluation 

Based on the results above, three active compounds (kaempferol, quercetin, and stigmasterol) and three target proteins (AKT1, IL-6, and TNF) were selected for molecular docking. The docking scores of the active compound to the target proteins are presented in the table (Table 3). The best docking image between the active compounds and the target proteins is visualized (Figure 7). The molecular docking results revealed that the binding energy of the effective compound with the target protein was nearly always less than or equal to -5 kcal/mol, indicating a strong binding effect. Additionally, the lower the binding energy between the compound and the target protein, the more favorable the binding. As demonstrated in this study, based on the binding energy, the binding ability of AKT1 to small molecules follows the order of stigmasterol > kaempferol > quercetin.

### 3.8. Molecular Dynamics Simulation Evaluation

#### 3.8.1. AKT1 with Kaempferol

The RMSD diagram indicates that the fluctuations of the protein and ligand are within an acceptable range. Particularly, starting from around 65 nanoseconds (ns), the fluctuation range of the protein’s Cα atom and ligand’s weight atom is within 3 Å. In terms of the RMS, the interaction between the protein residues and ligands is less variable compared to the other protein regions. Both of these aspects suggest that the complex is stable. The protein–ligand contact diagram reveals that multiple interactions exist between the protein and ligand, including hydrogen bonds, hydrophobic interactions, ion interactions, and water bridges. Residues such as TRP73, LYS193, and CYS196 interact with ligands to form hydrogen bonds, and there are hydrophobic interactions between VAL69, LEU168, VAL197, and the ligands. The small molecule compound is stable with the protein residues LYS193, CYS196, and VAL197 (Figure 8A).

#### 3.8.2. AKT1 with Quercetin

The RMSD diagram indicates that the fluctuations of the protein and ligand are within the acceptable range. Especially after about 70 ns, the fluctuation range of the protein Cα atom and ligand weight atom decreases. In the RMSF, the interaction between the protein residues and ligands is less volatile than in other protein regions. Both of these suggest that the complex is stable. The protein–ligand contact map shows that there are several interactions between the proteins and ligands, such as hydrogen bonds, hydrophobic interactions, and water bridges. Residues like TRP73, LYS193, and CYS196 interact with ligands to form hydrogen bonds, and there are hydrophobic interactions between VAL65, VAL69, LEU168, VAL171, PHE175, VAL197, and the ligands. The small molecule compounds are stable with the protein residues VAL69, VAL171, LYS193, CYS196, and VAL197 (Figure 8B).

#### 3.8.3. AKT1 with Stigmasterol

The RMSD diagram shows that the fluctuations of the protein and ligand are within the acceptable range. Especially starting from about 55 ns, the fluctuation range of the protein Cα atom and ligand weight atom decreases. In the RMSF, the interaction between the protein residues and ligands is less volatile than in other protein regions. Both of these aspects suggest that the complex is stable. The protein–ligand contact map shows that there are several interactions between the proteins and ligands, such as hydrogen bonds, hydrophobic interactions, and water bridges. The PHE269 residue interacts with the ligand to form hydrogen bonds, and there are hydrophobic interactions between VAL197, PHE208, PHE276, and the ligand. The small molecule compound is stable with the protein residues PHE269 and PHE276 (Figure 8C).

#### 3.8.4. IL-6 with Kaempferol

The RMSD diagram reveals that during the 100 ns process, the fluctuation range of the protein and ligand is large, suggesting that the protein might undergo significant conformational changes. At the end of 100 ns, the system has not reached equilibrium, and the simulation time might not be sufficient for a rigorous analysis. However, in the RMSF, the fluctuation of the interaction between the protein residues and ligands is smaller than that of other protein regions, indicating the stability of the complex. The protein–ligand contact diagram shows several interactions between the protein and ligand, such as hydrogen bonds, hydrophobic interactions, ion interactions, and water bridges. Residues GLN9, GLN147, and SER156 interact with ligands to form hydrogen bonds, and there are hydrophobic interactions between LYS126, PRO145, and the ligands. The small A molecule compound is stable with the protein residue PRO145 (Figure 9A).

#### 3.8.5. IL-6 with Quercetin

The RMSD diagram shows that the fluctuations of the protein and ligand are within an acceptable range. Especially from around 80 ns, the fluctuation range of the protein Cα atom and the ligand weight atom is within 3Å. In the RMSF, the fluctuation of the interaction between the protein residues and ligands is smaller than that of other protein regions. Both of these indicate that the complex is stable. The protein–ligand contact diagram shows that there are several interactions between the protein and ligand, including hydrogen bonds, hydrophobic interactions, ion interactions, and water bridges. Residues ALA127, GLN143, CYS146, and TYR148 interact with the ligands to form hydrogen bonds, and there are hydrophobic interactions between LYS126, PRO145, VAL128, and the ligands. The small molecule compound is stable with the protein residue THR125 (Figure 9B).

#### 3.8.6. IL-6 with Stigmasterol

The RMSD diagram shows that the fluctuations of the protein and the ligand are within an acceptable range. Especially from around 80 ns, the fluctuation range of the protein Cα atom and the ligand weight atom decreases. Moreover, the fluctuation of the interaction between the protein residues and ligands in the RMSF is small. Both of these indicate that the complex is stable. The protein–ligand contact map shows that there are several interactions between the proteins and ligands, such as hydrogen bonds, hydrophobic interactions, ion interactions, and water bridges. Residues ASN110, GLU144, and GLU163 interact with the ligands to form hydrogen bonds, and there are hydrophobic interactions between LEU108, PHE229, TYR230, and the ligands (Figure 9C).

#### 3.8.7. TNF with Stigmasterol

The RMSD diagram shows that the fluctuations of the protein and ligand are within an acceptable range. Especially from about 20 ns, the fluctuation range of the protein Cα atom and the ligand weight atom is within 3 Å. In the RMSF, most of the protein residues interact with the ligand, and the fluctuation range is smaller than that of other protein regions. Both of these indicate that the complex is stable. The protein–ligand contact map shows that there are several interactions between the proteins and ligands, such as hydrogen bonds, hydrophobic interactions, and water bridges. GLY121 and GLU135 residues interact with the ligands to form hydrogen bonds, and there are hydrophobic interactions between PHE64, TRP114, TYR141, and the ligands. The small molecule compound is stable with the protein residues PHE64 and GLU135 (Figure 10A).

#### 3.8.8. TNF with Quercetin

The RMSD diagram indicates that the fluctuations of the protein and the ligand are within an acceptable range. Especially from around 40 ns, the fluctuation range of the protein Cα atom and the ligand weight atom is within 3 Å. In the RMSF, the fluctuation of the interaction between the protein residues and ligands is smaller than that of other protein regions. Both of these suggest that the complex is stable. The protein–ligand contact map shows that there are several interactions between the proteins and ligands, including hydrogen bonds, hydrophobic interactions, ion interactions, and water bridges. GLU127 and ASP130 residues interact with the ligands to form hydrogen bonds, and there are hydrophobic interactions between ILE58, PHE124, LEU126, LEU132, and the ligands. The small molecule compound is stable with the protein residue ASP130 (Figure 10B).

#### 3.8.9. TNF with Kaempferol

The RMSD diagram reveals that the fluctuation of protein and ligands is within the acceptable range. Specifically, from around 10 ns, the fluctuation range of the protein Cα atom and the ligand weight atom is within 3 Å, but from about 90 ns, the fluctuation range becomes large, suggesting that the protein undergoes significant conformational changes. In the RMSF, most of the protein residues interact with the ligand, and the fluctuation range is smaller than that of other protein regions, both of which imply that the complex is stable. The protein–ligand contact diagram shows that there are several interactions between the protein and the ligand, such as hydrogen bonds, hydrophobic interactions, ion interactions, and water bridges. Residues T RP28, ASN46, GLN47, and ALA134 interact with the ligands to form hydrogen bonds, and there are hydrophobic interactions between LYS90, ILE136, and the ligands. The small molecule compounds are stable with the protein residues TRP28, ASN46, ALA134, and ILE136 (Figure 10C).

## 4. Discussion

The main objective of this study is to explore the underlying mechanism by which the JPYF II Formula treats COPD. By searching relevant databases and using active compound–target interaction network diagrams, the top three active compounds, namely quercetin, kaempferol, and stigmasterol, were selected. Pharmacological experiments have demonstrated that quercetin has anti-inflammatory, anti-cancer, and other effects [45,46,47]. Furthermore, relevant literature indicates that quercetin is a potential therapeutic agent for the prevention of COPD [48]. Kaempferol is a natural flavonol with potent anti-inflammatory properties [49,50], and experiments have shown that kaempferol can be used as a drug candidate for the treatment of COPD [51]. Stigmasterol, a plant sterol found in various herbs, has garnered significant attention due to its anti-inflammatory, antioxidant, and other pharmacological properties [52,53,54], and it can reduce peribronchial, perivascular, and alveolar infiltrates of inflammatory cells [55]. Consequently, quercetin, kaempferol, and stigmasterol are most likely considered the principal bioactive compounds of the JPYF II Formula in the treatment of COPD. A total of 156 common gene targets for COPD and the JPYF II Formula are considered potential gene targets for JPYF II Formula therapy for COPD. The PPI network diagram showed that AKT1, IL-6, and TNF were core targets. Serine/Threonine Kinase (AKT) is a versatile kinase that affects survival, proliferation, gene expression, and migration in most cell lines. AKT consists of three subtypes: AKT1, AKT2, and AKT3. AKT1 is mainly distributed in the heart, brain, and lungs. Research has indicated that AKT1 is involved in mediating the occurrence of inflammation [56,57,58]. Pharmacological experiments have demonstrated that inactivating the p16 aging pathway can prevent COPD by activating AKT [59]. Interleukin-6 (IL-6) is a pleiotropic cytokine with multiple functions in the body. Its crucial role in inflammatory diseases has been described in many reports [60,61,62]. Studies have shown that IL-6 is closely related to the development of COPD [63,64]. Tumor necrosis factor (TNF) consists of 19 ligands and 29 receptors. The regulatory effect of TNF on cells affects cell differentiation, survival, and related inflammatory responses. At the same time, TNF is a well-known pro-inflammatory cytokine that mediates inflammation [65,66,67,68]. TNF has been linked to the risk of COPD morbidity [69,70]. In summary, AKT1, IL-6, and TNF are likely to be targets for the JPYF II Formula in the treatment of COPD. 

To probe the JPYF II Formula mechanism in COPD therapy, we also performed GO and KEGG pathway enrichment analyses. The GO enrichment analysis revealed that a large number of target gene proteins were enriched for the biological function of inflammatory response. The results of the KEGG enrichment analysis indicated that a large number of target gene proteins were enriched in the PI3K/AKT signaling pathway. Therefore, it was predicted that this pathway would be the main signaling pathway for the treatment of COPD by using the JPYF II Formula. Phosphatidylinositol 3-kinase/protein kinase B (PI3K/AKT) mediates enzyme biological effects, including cell proliferation, apoptosis inhibition, etc. Studies have reported that the PI3K/AKT signaling pathway can inhibit the activation and migration of inflammatory cells and the production of cytokines. It is also a key signaling node in the fibrogenesis process, thus preventing the development of COPD [71,72,73]. Therefore, the treatment of COPD by using the JPYF II Formula may be through the PI3K/AKT signaling pathway to inhibit the occurrence of inflammation.

To probe the underlying molecular mechanism of the JPYF II Formula in COPD therapy, the screening results of the above network pharmacology were used, and the results were three active compounds and three target genes proteins. The three compounds, kaempferol, quercetin, and stigmasterol, are molecularly docked with AKT1, IL-6, and TNF, respectively. Through molecular docking, we verified the accuracy of the network pharmacology predictions. The results of molecular docking showed that the three compounds could bind well to the three target gene proteins, and the binding energy of the docking was less than 0, indicating that the three compounds had strong binding abilities and stability. In order to further understand the stability of the compound and the target gene protein set, MD simulations were performed. The results showed that the complex formed by TNF and the three compounds was more stable. This indicates that TNF may be a target for these compounds to exert their effects.

Although the latent compounds and target gene proteins for the treatment of COPD by using the JPYF II Formula have been identified through MD simulation, molecular docking, and network pharmacology, this study still has limitations. The accuracy and completeness of the database still need to be improved. Moreover, Chinese medicine components contain a large number of small molecule compounds, and the huge workload makes it impossible to evaluate COPD individually. At present, only a few active compounds can be obtained using the screening criteria for screening data. These limitations lead to the neglect of other parts of the compounds, resulting in incomplete results. Therefore, even if the three active compounds, quercetin, kaempferol, and stigmasterol, are considered to be the main components of the JPYF II Formula for the treatment of COPD, they still cannot represent all components of the JPYF II Formula. Therefore, in vitro and in vivo experiments must be used to verify the effects of other compound components on COPD. In addition, our research has not been covered before. Still, it provides more potential compounds, related target gene proteins, and signaling pathways for the treatment of COPD in the future, which has development space and practical significance. Therefore, the potential mechanism by which the JPYF II Formula treats COPD might involve quercetin, kaempferol, and stigmasterol acting on TNF, downregulating the PI3K/AKT signaling pathway, and thus achieving the effect of inhibiting the inflammatory response. The abbreviations used in this study are shown in Table 4.

## 5. Conclusions

This study employed molecular dynamics simulation, molecular docking, and network pharmacology methods to investigate the effective compounds, target gene proteins, and signaling pathways of the JPYF II Formula for treating COPD. The analyses revealed that kaempferol, quercetin, and stigmasterol are the main active compounds of the JPYF II Formula used for COPD treatment, and AKT1, IL-6, and TNF are the potential target gene proteins. It may mediate the biological function of the inflammatory response through the PI3K/AKT signaling pathway and play a role in the treatment of COPD. Additionally, this study provides new potential compounds and targets for subsequent COPD treatment.

## Figures and Tables

**Figure 1 biology-13-00711-f001:**
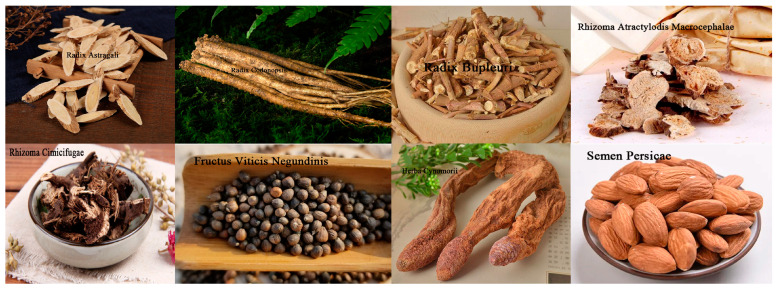
Eight traditional Chinese medicine ingredients in JPYF II granules.

**Figure 2 biology-13-00711-f002:**
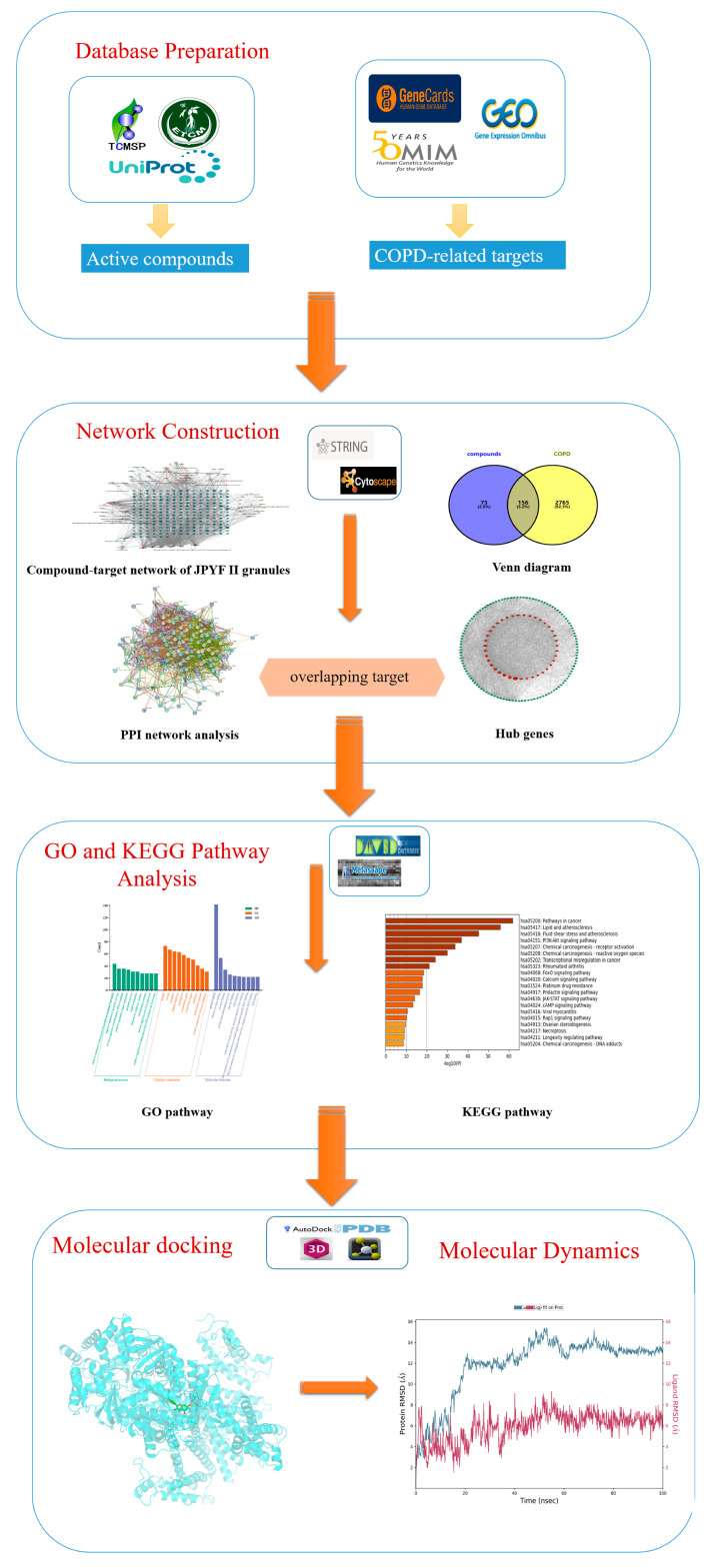
The workflow shows the mechanism of using network pharmacology for the treatment of COPD using JPYF II granules.

**Figure 3 biology-13-00711-f003:**
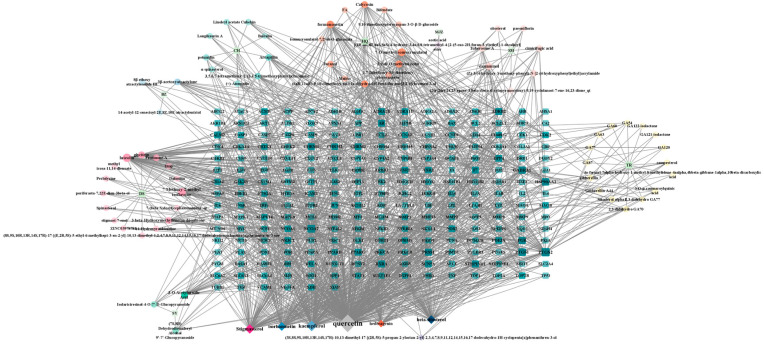
Compound–target network of JPYF II granules. The intermediate matrix stands for the target gene proteins related to JPYF II granules. The 6 diamonds represent the common active compounds among the ingredients. In the remaining part, the center of the circle represents the ingredient, and the circle represents the active compounds.

**Figure 4 biology-13-00711-f004:**
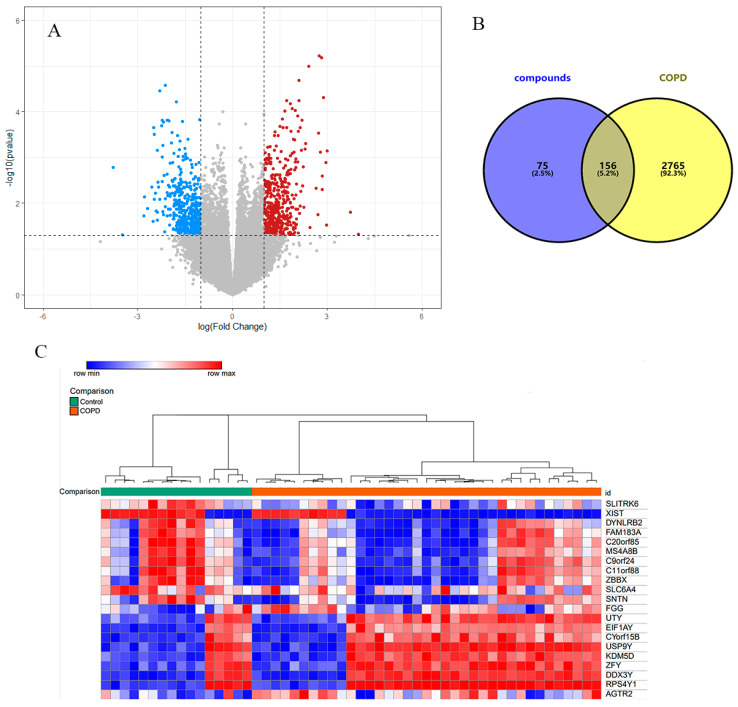
(**A**) Volcano plot of DEGs. (**B**) Venn diagram of the target of COPD and the target of JPYF II granules. (**C**) Heatmaps of partial gene expression between COPD and normal group.

**Figure 5 biology-13-00711-f005:**
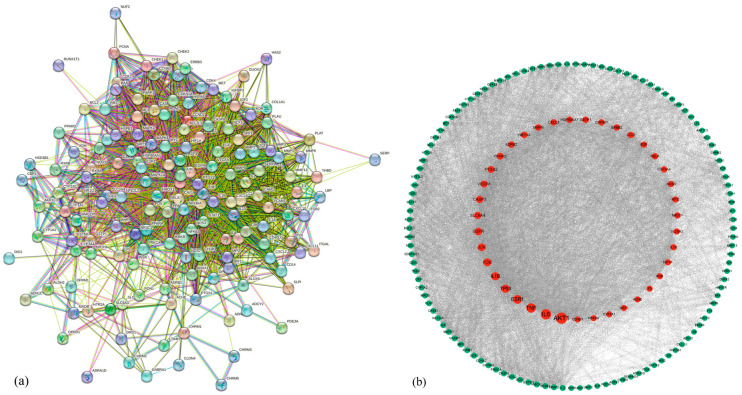
(**a**): The PPI network of COPD and JPYF II granules target. (**b**): Hub genes of COPD and JPYF II granules. (The larger the red dot, the more important the gene).

**Figure 6 biology-13-00711-f006:**
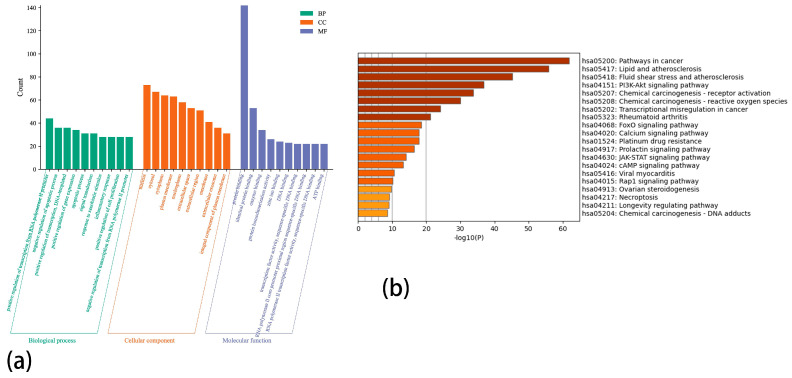
(**a**): Top 10 GO terms of hub genes. (**b**): Top 20 KEGG pathways of hub genes.

**Figure 7 biology-13-00711-f007:**
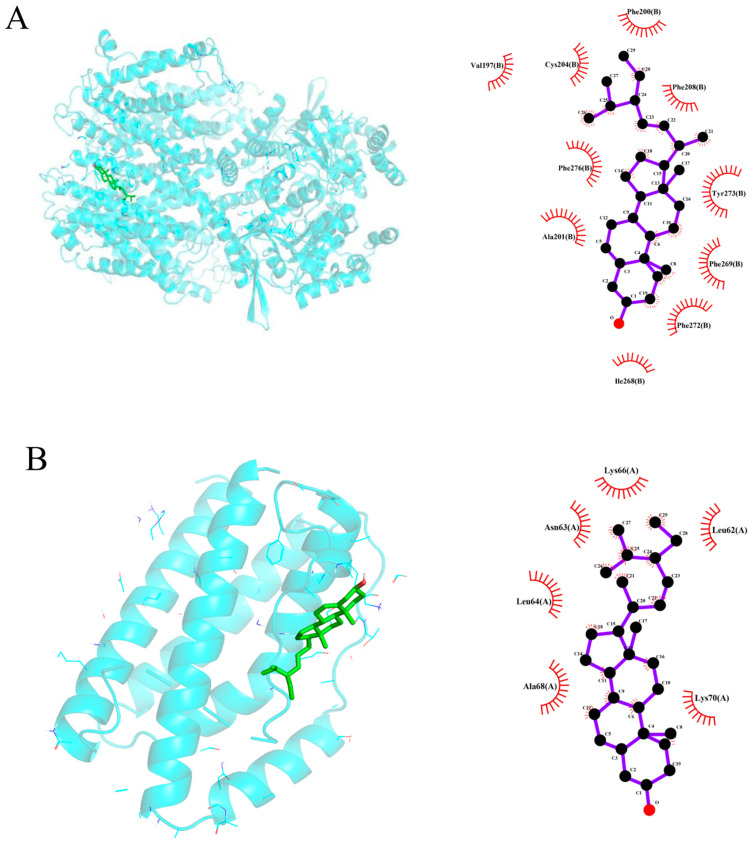
Molecular docking results of main active compounds of JPYF II granules: (**A**) stigmasterol-AKT1; (**B**) stigmasterol-IL-6; (**C**) stigmasterol-TNF; (**D**) quercetin-AKT1; (**E**) quercetin-IL-6; (**F**) quercetin-TNF; (**G**) kaempferol-AKT1; (**H**) kaempferol-IL-6; (**I**) kaempferol-TNF.

**Figure 8 biology-13-00711-f008:**
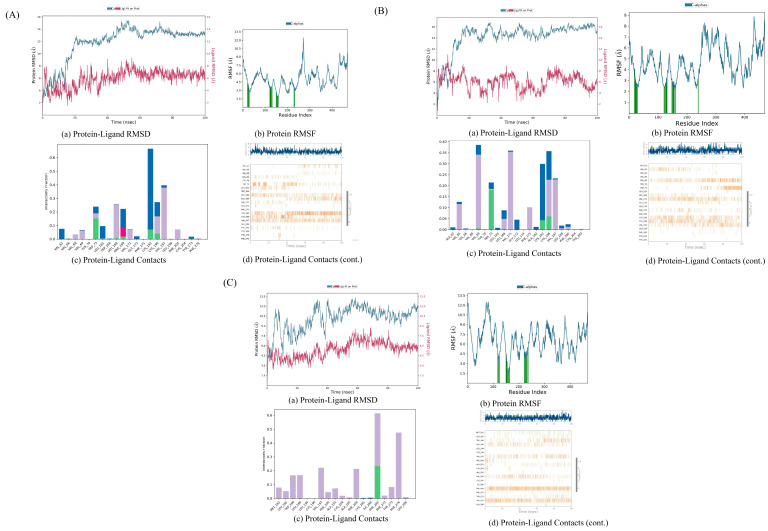
MD simulation result of complex structure. (**A**) AKT1- kaempferol; (**B**) AKT1- quercetin; (**C**) AKT1-stigmasterol.

**Figure 9 biology-13-00711-f009:**
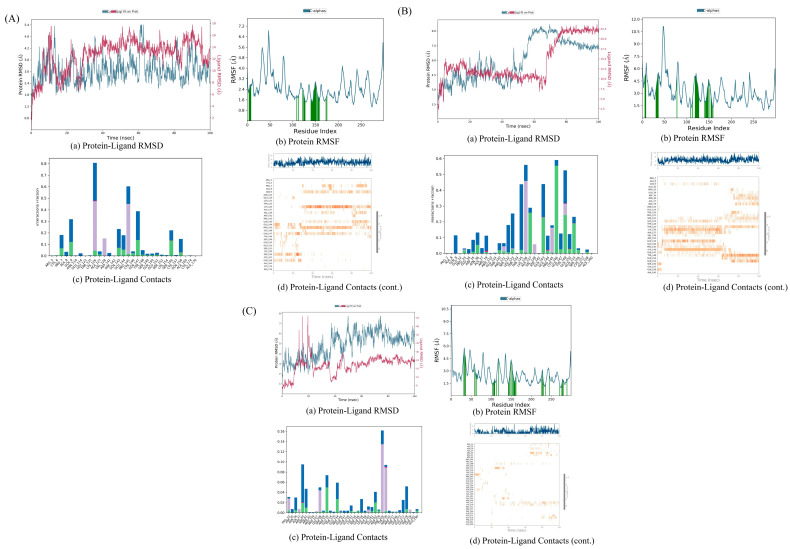
MD simulation result of complex structure: (**A**) IL-6-kaempferol; (**B**) IL-6-quercetin; (**C**) IL-6-stigmasterol.

**Figure 10 biology-13-00711-f010:**
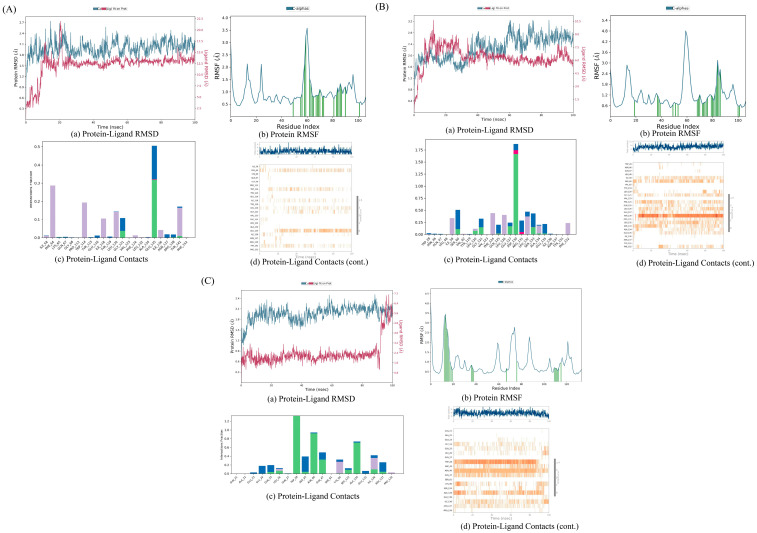
MD simulation result of complex structure: (**A**) TNF-stigmasterol; (**B**) TNF-quercetin; (**C**) TNF-kaempferol.

**Table 1 biology-13-00711-t001:** The common active compounds between the ingredients.

Molecule Name	OB (%)	DL	Origin
Stigmasterol	43.83	0.76	CH, DS, SM
Isorhamnetin	49.6	0.31	CH, HQ
Kaempferol	41.88	0.24	CH, HQ
Quercetin	46.43	28	CH, HQ
Hederagenin	36.91	0.75	HQ, TR
(3S,8S,9S,10R,13R,14S,17R)-10,13-dimethyl-17-[(2R,5S)-5-propan-2-yloctan-2-yl]-2,3,4,7,8,9,11,12,14,15,16,17-dodecahydro-1H-cyclopenta[a]phenanthren-3-ol	36.23	0.78	HQ, BZ
Beta-sitosterol	36.91	0.75	MJZ, TR

**Table 2 biology-13-00711-t002:** Top three compounds’ information for JPYF II Formula network.

Compound	Molecule Structure	Degree	Average Shortest PathLength	Betweenness Centrality	Closeness Centrality
quercetin	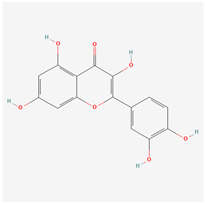	292	1.894230769	0.514216076	0.527918782
kaempferol	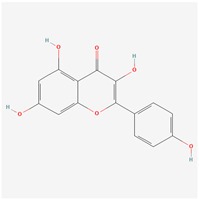	116	2.432692308	0.101388715	0.411067194
Stigmasterol	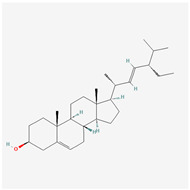	84	2.618589744	0.050442562	0.381884945

**Table 3 biology-13-00711-t003:** The binding energy of compound and core targets (kcal/mol).

Target	Target (PDB ID)	Target Structure	Compound	Affinity (kcal/mol)
AKTI	7WM2	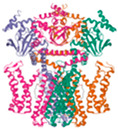	quercetin	−7.6
			kaempferol	−8.8
			stigmasterol	−8.9
IL-6	1ALU	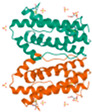	quercetin	−7.1
			kaempferol	−6.7
			stigmasterol	−4.3
TNF	5UUI	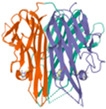	quercetin	−6.9
			kaempferol	−6.7
			stigmasterol	−5.9

**Table 4 biology-13-00711-t004:** Abbreviations of this study.

Full Name	Abbreviations
JianPiYiFei II granules	JPYF II granules
Chronic obstructive pulmonary disease	COPD
molecular dynamics	MD
Radix Astragali	HQ
Radix Codonopsis	DS
Radix Bupleuri	CH
Rhizoma Atractylodis Macrocephalae	BZ
Rhizoma Cimicifugae	SM
Fructus Viticis Negundinis	MJZ
Herba Cynomorii	SY
Semen Persicae	TR
Traditional Chinese medicine	TCM
Traditional Chinese Medicine Systems Pharmacology Database	TCMSP
Encyclopedia of Traditional Chinese Medicine	ETCM
Drug similarity	DL
Oral bioavailability	OB
Gene Expression Omnibus	GEO
Differentially expressed genes	DEGs
Online Mendelian Inheritance in Man	OMIM
Protein–protein interaction	PPI
Betweenness centrality	BC
Biological processes	BP
Cellular components	CC
Molecular function	MF
Protein Data Bank	PDB
Interleukin-6	IL-6
Tumor necrosis factor	TNF

## Data Availability

The data from this study are included in the article. Other outcome inquiries can be communicated with the corresponding authors.

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
