# Peer review of "Mechanisms Underlying the Therapeutic Effects of JianPiYiFei II Granules in Treating COPD Based on GEO Datasets, Network Pharmacology, Molecular Docking, and Molecular Dynamics Simulations"

_biology, 2024, doi:10.3390/biology13090711_

Round 1

Reviewer 1 Report

Comments and Suggestions for Authors

In this study, the authors elucidate the effective compounds, targets, and related molecular mechanisms of JPYF II granules, Chinese medicine for the treatment of Chronic obstructive pulmonary disease (COPD), through network pharmacology, molecular docking, and MD simulation techniques. They found that AKT1, IL-6, and TNF were key target gene proteins of JPYF II. These methodologies provide a new strategy to identify the molecular mechanism of effective compounds. Also, understanding the mechanisms of JPYF II in treating COPD is significant. Thus, it might have a high impact on the field. However, a few weaknesses need to be addressed.

1.       To understand if the strategies really work, it is essential to confirm whether AKT1, IL-6, and TNF were indeed key target gene proteins of JPYF II. Does JPYF II affect the expression or activity of these proteins in the pulmonary cells or tissue? The results would significantly strengthen the premise and potential impact.

2.       Some figures, e.g., Figs 2, 4, and 5, are too small to be seen.

Author Response

Comment 1:

To understand if the strategies really work, it is essential to confirm whether AKT1, IL-6, and TNF were indeed key target gene proteins of JPYF II. Does JPYF II affect the expression or activity of these proteins in the pulmonary cells or tissue? The results would significantly strengthen the premise and potential impact.

Response 1:

Thank you for your suggestion. The question you mentioned is the limitation of this study, which we have focused on in the discussion section. For the limitations of this study, we will carry out subsequent experiments to further explore its potential mechanism.

Comment 2:

Some figures, e.g., Figs 2, 4, and 5, are too small to be seen.

Response 2:

Thank you for your suggestion, we have revised it in the latest draft.

Reviewer 2 Report

Comments and Suggestions for Authors

This manuscript used the database, program pathway, and simulation to study the mechanisms for COPD by JPYF II.

The main issue of the current manuscript is the lack of experimental data to support the simulation and dataset analysis. There is a concern that lack of experimental data to really support if those 3 main ingredients are the real main 3 ingredients. I would suggest that the title should add one word "potential" before "mechanisms underlying ....". Without experimental data, it is very difficult to decide if the proposed mechanisms are close to real things. Reference citing is fine. As I mentioned on my report, what would be the main limitation of the current study?

Comments:

1. Please list all the abbreviations.

2. Why not eliminate each compound to evaluate the effect on COPD?

3. Please show 8 ingredients' images.

4. On Table 2: Is the data predicted? Any chemical purification to support?

5. What would be the main limitation of the current study?

Author Response

Comments 1:

Please list all the abbreviations.

Response 1:

Thank you for your suggestion, we have revised it in the latest draft.

Comments 2:

Why not eliminate each compound to evaluate the effect on COPD?

Response 2:

Thank you for your question. Because traditional Chinese medicine contains a large number of small molecule compounds, COPD cannot be evaluated one by one. Only the top small molecule compounds can be studied first to explore their role in the treatment of COPD, so this is the limitation of this study.

Comments 3: 

Please show 8 ingredients' images

Response 3:

Thank you for your suggestion, we have revised it in the latest draft.

Comments 4: 

On Table 2: Is the data predicted? Any chemical purification to support?

Response 4:

Thank you for your questions, the main parameter is the degree value, and the references are cited in the latest manuscripts

Comments 5: 

What would be the main limitation of the current study?

Response 5:

Thank you for your questions, the limitations of this study are highlighted in the discussion section

Reviewer 3 Report

Comments and Suggestions for Authors

1. Figure 2 can not understand clearly. Make it more clear and the bottom indications in color also not visible.

2. Table.2  what parameters are used inthis. There was no information.

3. Abstarct: for what aim the work?. What achieved should be need to indicate.

4. Am not seen any key highlight of this work. How it merit over other ?

5. Try to highlight and compare this work wih previous works.

Additional comments:

1. At 3.7 sub head. " lower the binding energy of the compound to the target protein, the better the binding" can you explain this using one example from this research.

2. Is there any previous reports that have the similar type of molecules that shown any other biological activity? If so discuss at introduction.

3. References not appropriate: For example at introduction part. "According to related reports, COPD causes high mortality and is predicted to become the third leading cause of death in the world by 2020." This statement valid for current year ? And no reference cited. And Also one more next to this "Therefore, COPD is a health care issue of concern[3, 4]" ? What does it mean? and References ?.

Comments on the Quality of English Language

Minor editing needed 

Author Response

Comments 1:

Figure 2 can not understand clearly. Make it more clear and the bottom indications in color also not visible

Response 1:

Thank you for your suggestion, we have revised it in the latest draft.

Comments 2:

Table.2  what parameters are used inthis. There was no information.

Response 2:

Thank you for your questions, the main parameter is the degree value, and the references are cited in the latest manuscripts

Comments 3:

Abstarct: for what aim the work?. What achieved should be need to indicate.

Response 3:

Thank you for your questions, the specific elaboration of the update in the latest manuscript

Comments 4:

Am not seen any key highlight of this work. How it merit over other ?

Response 4:

Thank you for your question, using the current trend of means to explore the potential mechanism of the treatment of COPD, to provide new ideas for further treatment of COPD.

Comments 5:

Try to highlight and compare this work wih previous works.

Response 5:

Thank you for your question. Compared with the past, this method can save a lot of time to explore the treatment mechanism and clarify a clearer direction.

Additional comments 1:

At 3.7 sub head. " lower the binding energy of the compound to the target protein, the better the binding" can you explain this using one example from this research.

Response 1:

Thank you for your suggestion, we have revised it in the latest draft.

Additional comments 2:

 Is there any previous reports that have the similar type of molecules that shown any other biological activity? If so discuss at introduction.

Response 2:

Thank you for your question, we have revised it in the latest draft.

Additional comments 3:

 References not appropriate: For example at introduction part. "According to related reports, COPD causes high mortality and is predicted to become the third leading cause of death in the world by 2020." This statement valid for current year ? And no reference cited. And Also one more next to this "Therefore, COPD is a health care issue of concern[3, 4]" ? What does it mean? and References ?.

Response 3:

Thank you for your questions and suggestions, we have modified the references cited, and some statements have been modified and updated in the latest manuscripts.

Reviewer 4 Report

Comments and Suggestions for Authors

Dear authors,

I read with great interest the article by Liyuan Pang et al. “Mechanisms underlying the therapeutic effects of JPYF II granules in treating COPD based on GEO datasets, network pharmacology, molecular docking and molecular dynamics simulation”.

Understanding the mechanisms of action of substances is an extremely important aspect of fundamental biology and has practical significance. The task set by the authors in this work has been accomplished. The authors identified the main active compounds of JPYF II Formula (kaempferol, quercetin, stigmasterol) and the potential target genes protein for JPYF II Formula (AKT1, IL-6, TNF). All this expands our knowledge and capabilities in the treatment of COPD. The methods used are modern and correspond to the tasks set. This work corresponds to the subject of the journal.

The manuscript is well written, the illustrations are good, but could be enlarged to improve the presentation. All this allows me to recommend this manuscript for publication in the journal in its current form without significant changes.

With respect.

Author Response

Thank you for suggesting that we will further explore the potential mechanisms for the treatment of COPD through experimental verification and other means.

Round 2

Reviewer 1 Report

Comments and Suggestions for Authors

My concerns have been addressed.

Reviewer 2 Report

Comments and Suggestions for Authors

No more comments.